# Sparse PCA from Sparse Linear Regression

**Guy Bresler**
MIT
guy@mit.edu

**Sung Min Park**
MIT
sp765@mit.edu

**Mădălina Persu**
Two Sigma,[*] MIT
mpersu@mit.edu

## Abstract

Sparse Principal Component Analysis (SPCA) and Sparse Linear Regression (SLR) have a wide range of applications and have attracted a tremendous amount of attention in the last two decades as canonical examples of statistical problems in high dimension. A variety of algorithms have been proposed for both SPCA and SLR, but an explicit connection between the two had not been made. We show how to efficiently transform a black-box solver for SLR into an algorithm for SPCA: assuming the SLR solver satisfies prediction error guarantees achieved by existing efficient algorithms such as those based on the Lasso, the SPCA algorithm derived from it achieves near state of the art guarantees for testing and for support recovery for the single spiked covariance model as obtained by the current best polynomial-time algorithms. Our reduction not only highlights the inherent similarity between the two problems, but also, from a practical standpoint, allows one to obtain a collection of algorithms for SPCA directly from known algorithms for SLR. We provide experimental results on simulated data comparing our proposed framework to other algorithms for SPCA.

## 1 Introduction

Principal component analysis (PCA) is a fundamental technique for dimension reduction used widely in data analysis. PCA projects data along a few directions that explain most of the variance of observed data. One can also view this as linearly transforming the original set of variables into a (smaller) set of uncorrelated variables called principal components.

Recent work in high-dimensional statistics has focused on sparse principal component analysis (SPCA), as ordinary PCA estimates become inconsistent in this regime [22]. In SPCA, we restrict the principal components to be sparse, meaning they have only a few nonzero entries in the original basis. This has the advantage, among others, that the components are more interpretable [23, 49], while components may no longer be uncorrelated. We study SPCA under the Gaussian *(single) spiked covariance model* introduced by [21]: we observe $n$ samples of a random variable $X$ distributed according to a Gaussian distribution $\mathcal{N}(0, I_d + \theta uu^\top)$, where $||u||_2 = 1$ with at most $k$ nonzero entries.[2] $I_d$ is the $d \times d$ identity matrix, and $\theta$ is the signal-to-noise parameter. We study two settings of the problem, hypothesis testing and support recovery.

Sparsity assumptions have played an important role in a variety of other problems in high-dimensional statistics, in particular linear regression. Linear regression is also ill-posed in high dimensions, so via imposing sparsity on the regression vector we recover tractability.

Though the literature on two problems are largely disjoint, there is a striking similarity between the two problems, in particular when we consider statistical and computational trade-offs. The

---

[*]The views expressed herein are solely the views of the author(s) and are not necessarily the views of Two Sigma Investments, LP or any of its affiliates. They are not intended to provide, and should not be relied upon for, investment advice.

[2]Sometimes we will write this latter condition as $u \in B_0(k)$ where $B_0(k)$ is the "$\ell_0$-ball" of radius $k$.

natural information-theoretically optimal algorithm for SPCA [4] involves searching over all possible supports of the hidden spike. This bears resemblance to the minimax optimal algorithm for SLR [35], which optimizes over all sparse supports of the regression vector. Both problems appear to exhibit gaps between statistically optimal algorithms and known computationally efficient algorithms, and conditioned on relatively standard complexity assumptions, these gaps seem irremovable [3, 44, 47].

## 1.1 Our contributions

In this paper we give algorithmic evidence that this similarity is likely not a coincidence. Specifically, we give a simple, general, and efficient procedure for transforming a black-box solver for sparse linear regression to an algorithm for SPCA. At a high level, our algorithm tries to predict each coordinate[3] linearly from the rest of the coordinates using a black-box algorithm for SLR. The advantages of such a black-box framework are two fold: theoretically, it highlights a structural connection between the two problems; practically, it allows one to simply plug in any of the vast majority of solvers available for SLR and directly get an algorithm for SPCA with provable guarantees. In particular,

- For hypothesis testing: we match state of the art provable guarantee for computationally efficient algorithms; our algorithm successfully distinguishes between isotropic and spiked Gaussian distributions at signal strength $\theta \gtrsim \sqrt{\frac{k^2 \log d}{n}}$. This matches the phase transition of diagonal thresholding [22] and Minimal Dual Perturbation [4] up to constant factors.

- For support recovery: for general $p$ and $n$, when each non-zero entry of $u$ is at least $\Omega(1/\sqrt{k})$ (a standard assumption in the literature), our algorithm succeeds with high probability for signal strength $\theta \gtrsim \sqrt{\frac{k^2 \log d}{n}}$, which is nearly optimal.[4]

- In experiments, we demonstrate that using popular existing SLR algorithms as our black-box results in reasonable performance.

- We theoretically and empirically illustrate that our SPCA algorithm is also robust to *rescaling* of the data, for instance by using a Pearson correlation matrix instead of a covariance matrix.[5] Many iterative methods rely on initialization via first running diagonal thresholding, which filters variables with higher variance; rescaling renders diaganoal thresholding useless, so in some sense our framework is more robust.

## 2 Preliminaries

### 2.1 Problem formulation for SPCA

**Hypothesis testing** Here, we want to distinguish whether $X$ is distributed according to an isotropic Gaussian or a spiked covariance model. That is, our null and alternate hypotheses are:

$$H_0 : X \sim \mathcal{N}(0, I_d) \text{ and } H_1 : X \sim \mathcal{N}(0, I_d + \theta u u^\top),$$

Our goal is to design a test $\psi \colon \mathbb{R}^{n \times d} \to \{0, 1\}$ that discriminates $H_0$ and $H_1$. More precisely, we say that $\psi$ discriminates between $H_0$ and $H_1$ with probability $1 - \delta$ if both type I and II errors have a probability smaller than $\delta$:

$$\mathbf{P}_{H_0}(\psi(X) = 1) \le \delta \text{ and } \mathbf{P}_{H_1}(\psi(X) = 0) \le \delta.$$

We assume the following additional condition on the spike $u$:

(C1) $c_{min}^2/k \le u_i^2 \le 1 - c_{min}^2/k$ for at least one $i \in [d]$ where $c_{min} > 0$ is some constant.

The above condition says that at least one coordinate has enough mass, yet the mass is not entirely concentrated on just that singlecoordinate. Trivially, we always have at least one $i \in [d]$ s.t. $u_i^2 \geq 1/k$, but this is not enough for our regression setup, since we want at least one other coordinate $j$ to have sufficient correlation with coordinate $i$. We remark that the above condition is a very mild technical condition. If it were violated, almost all of the mass of $u$ is on a single coordinate, so a simple procedure for testing the variance (which is akin to diagonal thresholding) would suffice.

**Support recovery** The goal of support recovery is to identify the support of $u$ from our samples $\mathbf{X}$. More precisely, we say that a support recovery algorithm succeeds if the recovered support $\widehat{S}$ is the same as $S$, the true support of $u$. As standard in the literature [1, 31], we need to assume a minimal bound on the size of the entries of $u$ in the support.

For our support recovery algorithm, we will assume the following condition (note that it implies Condition (C1) and is much stronger):

(C2)  $|u_i| \geq c_{min}/\sqrt{k}$ for some constant $0 < c_{min} < 1 \ \forall i \in [d]$

Though the settings are a bit different, this minimal bound along with our results are consistent with lower bounds known for sparse recovery. These lower bounds ([18, 42]; bound of [18] is a factor of $k$ weaker) imply that the number of samples must grow roughly as $n \gtrsim (1/u_{min}^2)k \log d$ where $u_{min}$ is the smallest entry of our signal $u$ normalized by $1/\sqrt{k}$, which is qualitativley the same threhold required by our theorems.

## 2.2 Background on SLR

In linear regression, we observe a response vector $y \in \mathbb{R}^n$ and a design matrix $\mathbb{X} \in \mathbb{R}^{n \times d}$ that are linked by the linear model $y = \mathbb{X}\beta^* + w$, where $w \in \mathbb{R}^n$ is some form of observation noise, typically with i.i.d. $\mathcal{N}(0, \sigma^2)$ entries. Our goal is to recover $\beta^*$ given noisy observations $y$. While the matrices $\mathbb{X}$ we consider arise from a (random) correlated design (as analyzed in [42], [43]), it will make no difference to assume the matrices are deterministic by conditioning, as long as the distribution of the design matrix and noise are independent, which we will demonstrate in our case. Most of the relevant results on sparse linear regression pertain to deterministic design.

In sparse linear regression, we additionally assume that $\beta^*$ has only $k$ non-zero entries, where $k \ll d$. This makes the problem well posed in the high-dimensional setting. Commonly used performance measures for SLR are tailored to prediction error ($1/n\|\mathbb{X}\beta^* - \mathbb{X}\widehat{\beta}\|_2^2$ where $\widehat{\beta}$ is our guess), support recovery (recovering support of $\beta^*$), or parameter estimation (minimizing $\|\beta^* - \widehat{\beta}\|$ under some norm). We focus on prediction error, analyzed over random realizations of the noise. There is a large amount of work on SLR and we defer a more in-depth overview to Appendix A.

Most efficient methods for SLR impose certain conditions on $\mathbb{X}$. We focus on the *restricted eigenvalue* condition, which roughly stated makes the prediction loss strongly convex near the optimum:

**Definition 2.1** (Restricted eigenvalue [47]). First define the cone $\mathbb{C}(S) = \{\beta \in \mathbb{R}^d \mid \|\beta_{S^c}\|_1 \leq 3\|\beta_S\|_1\}$, where $S^c$ denotes the complement, $\beta_T$ is $\beta$ restricted to the subset $T$. The *restricted eigenvalue* (RE) constant of $\mathbb{X}$, denoted $\gamma(\mathbb{X})$, is defined as the largest constant $\gamma > 0$ s.t.

$$1/n\|\mathbb{X}\beta\|_2^2 \geq \gamma\|\beta\|_2^2 \quad \text{for all } \beta \in \bigcup_{|S|=k, S \subseteq [d]} \mathbb{C}(S)$$

For more discussion on the restricted eigenvalue, see Appendix A.

**Black-box condition** Given the known guarantees on SLR, we define a condition that is natural to require on the guarantee of our SLR black-box, which is invoked as $\mathsf{SLR}(y, \mathbb{X}, k)$.

**Condition 2.2** (Black-box condition). Let $\gamma(\mathbb{X})$ denote the restricted eigenvalue of $\mathbb{X}$. There are universal constants $c, c', c''$ such that $\mathsf{SLR}(y, \mathbb{X}, k)$ outputs $\widehat{\beta}$ that is $k$-sparse and satisfies:

$$\frac{1}{n}\|\mathbb{X}\widehat{\beta} - \mathbb{X}\beta^*\|_2^2 \leq \frac{c}{\gamma(\mathbb{X})^2}\frac{(\sigma^2 k \log d)}{n} \quad \forall \beta^* \in B_0(k) \text{ w.p.} \geq 1 - c'\exp(-c''k\log d)$$

# 3 Algorithms and main results

We first discuss how to view samples from the spiked covariance model in terms of a linear model. We then give some intuition motivating our statistic. Finally, we state our algorithms and main theorems, and give a high-level sketch of the proof.

## 3.1 The linear model

Let $X^{(1)}, X^{(2)}, \ldots, X^{(n)}$ be $n$ i.i.d. samples from the spiked covariance model; denote as $\mathbf{X} \in \mathbb{R}^{n \times d}$ the matrix whose rows are $X^{(i)}$. Intuitively, if variable $i$ is contained in the support of the spike, then the rest of the support should allow to provide a nontrivial prediction for $\mathbf{X}_i$ since variables in the support are correlated. Conversely, for $i$ not in the support (or under the isotropic null hypothesis), all of the variables are independent and other variables are useless for predicting $\mathbf{X}_i$. So we regress $\mathbf{X}_i$ onto the rest of the variables.

Let $\mathbf{X}_{-i}$ denote the matrix of samples in the SPCA model with the $i$th column removed. For each column $i$, we can view our data as coming from a linear model with design matrix $\mathbb{X} = \mathbf{X}_{-i}$ and the response variable $y = \mathbf{X}_i$.

The "true" regression vector depends on $i$. Under the alternate hypothesis $H_1$, if $i \in S$, we can write $y = \mathbb{X}\beta^* + w$ where $\beta^* = \frac{\theta u_i}{1+(1-u_i^2)\theta} u_{-i}$ and $w \sim \mathcal{N}(0, \sigma^2)$ with $\sigma^2 = 1 + \frac{\theta u_i^2}{1+(1-u_i^2)\theta}$ [6] If $i \notin S$, and for any $i \in [d]$ under the null hypothesis, $y = w$ where $w = \mathbf{X}_i \sim \mathcal{N}(0,1)$ (implicitly $\beta^* = \mathbf{0}$).

## 3.2 Designing the test statistic

Based on the linear model above, we want to compute a test statistic that will indicate when a coordinate $i$ is on support. Intuitively, we predictive power of our linear model should be higher when $i$ is on support. Indeed, a calculation shows that the variance in $X_i$ is reduced by approximately $\theta^2/k$. We want to measure this reduction in noise to detect when $i$ is on support or not.

Suppose for instance that we have access to $\beta^*$ rather than $\widehat{\beta}$ (note that this is not possible in practice since we do not know the support!). Since we want to measure the reduction in noise when the variable is on support, as a first step we might try the following statistic:

$$Q_i = {}^{1}/n\|y - \mathbb{X}\beta^*\|_2^2$$

Unfortunately, this statistic will not be able to distinguish the two hypotheses, as the reduction in the above error is too small (on the order of $\theta^2/k$ compared to overall order of $1 + \theta$), so deviation due to random sampling will mask any reduction in noise. We can fix this by adding the variance term $\|y\|^2$:

$$Q_i = {}^{1}/n\|y\|_2^2 - {}^{1}/n\|y - \mathbb{X}\beta^*\|_2^2$$

On a more intuitive level, including $\|y\|_2^2$ allows us to measure the relative *gain* in predictive power without being penalized by a possibly large variance in $y$. Fluctuations in $y$ due to noise will typically be canceled out in the difference of terms in $Q_i$, minimizing the variance of our statistic.

We have to add one final fix to the above estimator. We obviously do not have access to $\beta^*$, so we must use the estimate $\widehat{\beta} = \mathsf{SLR}(y, \mathbb{X}, k)$ ($y, \mathbb{X}$ are as defined in Section 3.1) which we get from our black-box. As our analysis shows, this substitution does not affect much of the discriminative power of $Q_i$ as long as the SLR black-box satisfies prediction error guarantees stated in Condition 2.2. This gives our final statistic:[7]

$$Q_i = {}^{1}/n\|y\|_2^2 - {}^{1}/n\|y - \mathbb{X}\widehat{\beta}\|_2^2.$$

## 3.3 Algorithms

Below we give algorithms for hypothesis testing and for support recovery, based on the $Q$ statistic:

**Algorithm 1** $Q$-hypothesis testing

Input: $\mathbf{X} \in \mathbb{R}^{d \times n}, k$
Output: $\{0, 1\}$
**for** $i = 1, \ldots, d$ **do**
  $\widehat{\beta}_i = \mathsf{SLR}(\mathbf{X}_i, \mathbf{X}_{-i}, k)$
  $Q_i = \frac{1}{n}\|\mathbf{X}_i\|_2^2 - \frac{1}{n}\|\mathbf{X}_i - \mathbf{X}_{-i}\widehat{\beta}_i\|_2^2$
  **if** $Q_i > \frac{13k \log \frac{d}{k}}{n}$ **then**
    return 1
  **end if**
**end for**
Return 0

**Algorithm 2** $Q$-support recovery

Input: $\mathbf{X} \in \mathbb{R}^{d \times n}, k$
$\widehat{S} = \varnothing$
**for** $i = 1, \ldots, d$ **do**
  $\widehat{\beta}_i = \mathsf{SLR}(\mathbf{X}_i, \mathbf{X}_{-i}, k)$
  $Q_i = \frac{1}{n}\|\mathbf{X}_i\|_2^2 - \frac{1}{n}\|\mathbf{X}_i - \mathbf{X}_{-i}\widehat{\beta}_i\|_2^2$
  **if** $Q_i > \frac{13k \log \frac{d}{k}}{n}$ **then**
    $\widehat{S} \leftarrow \widehat{S} \cup \{i\}$
  **end if**
**end for**
Return $\widehat{S}$

Below we summarize our guarantees for the above algorithms. The proofs are simple, but we defer them to Appendix C.

**Theorem 3.1** (Hypothesis test). *Assume we have access to* $\mathsf{SLR}$ *that satisfies Condition 2.2 and with runtime* $T(d, n, k)$ *per instance. Under Condition (C1), there exist universal constants* $c_1, c_2, c_3, c_4$ *s.t. if* $\theta^2 > \frac{c_1}{c_{min}^2}\frac{k^2 \log d}{n}$ *and* $n > c_2 k \log d$, *Algorithm 1 outputs* $\psi$ *s.t.*

$$\mathbf{P}_{H_0}(\psi(X) = 1) \vee \mathbf{P}_{H_1}(\psi(X) = 0) \le c_3 \exp(-c_4 k \log d)$$

*in time* $O(dT + d^2 n)$.

**Theorem 3.2** (Support recovery). *Under the same conditions as above plus Condition (C2), if* $\theta^2 > \frac{c_1}{c_{min}^2}\frac{k^2 \log d}{n}$, *Algorithm 2 above finds* $\widehat{S} = S$ *with probability at least* $1 - c_3 \exp(-c_4 k \log d)$ *in time* $O(dT + d^2 n)$.

### 3.4 Comments

**RE for sample design matrix** Because population covariance $\Sigma = \mathbb{E}[\mathbb{X}\mathbb{X}^\top]$ has minimum eigenvalue 1, with high probability the sample design matrix $\mathbb{X}$ has constant restricted eigenvalue value given enough samples, i.e. $n$ is large enough (see Appendix B.3 for more details), and the prediction error guarantee of Condition 2.2 will be good enough for our analysis.

**Running time** The runtime of both Algorithm 1 and 2 is $\tilde{O}(nd^2)$. The discussion presented at the end of Appendix C details why this is competitive for (single spiked) SPCA, at least theoretically.

**Unknown sparsity** Throughout the paper we assume that the sparsity level $k$ is known. However, if $k$ is unknown, standard techniques could be used to adaptively find approximate values of $k$ ([16]). For instance, for hypothesis testing, we can start with an initial overestimate $k'$, and keep halving until we get enough coordinates $i$ with $Q_i$ that passes the threshold for the given $k'$.

**Robustness of Q statistic to rescaling**

Intuitively, our algorithms for detecting correlated structure in data should be invariant to rescaling of the data; the precise scale or units for which one variable is measured should not have an impact on our ability to find meaningful structure underlying the data. Our algorithms based on $Q$ are robust to rescaling, perhaps unsurprisingly, since correlations between variables in the support remain under rescaling.

On the other hand, diagonal thresholding, an often-used preprocessing step for SPCA which filters variables strictly based on variance, would trivially fail under rescaling. This illustrates a strength of our framework over other existing algorithms for SPCA.

Below we show explicitly that $Q$ statistics are indeed robust to rescaling: Let $\widetilde{X} = DX$ be the rescaling of $X$, where $D$ is some diagonal matrix. Let $D_S$ be $D$ restricted to rows and columns in $S$. Note that $\widetilde{\Sigma}$, the covariance matrix of the rescaled data, is just $D\Sigma D$ by expanding the definition.

Similarly, note $\widetilde{\Sigma}_{2:d,1} = D_1 D_{2:d}\Sigma_{2:d,1}$ where $D_{2:d}$ denotes $D$ without row and column 1. Now, recall the term which dominated our analysis of $Q_i$ under $H_1$, $(\beta^*)^\top\Sigma_{2:d}\beta^*$, which was equal to

$$\Sigma_{1,2:d}\Sigma_{2:d}^{-1}\Sigma_{2:d,1}$$

We replace the covariances by their rescaled versions to obtain:

$$\tilde{\beta}^{*\top}\widetilde{\Sigma}\tilde{\beta}^* = (D_1\Sigma_{1,2:d}D_{2:d})D_{2:d}^{-1}\Sigma_{2:d}^{-1}D_{2:d}^{-1}(D_{2:d}\Sigma_{2:d,1}D_1) = D_1^2(\beta^*)^\top\Sigma_{2:d}\beta^*$$

For the spiked covariance model, rescaling variances to one amount to rescaling with $D_1 = 1/(1+\theta)$. Thus, we see that our signal strength is affected only by constant factor (assuming $\theta \leq 1$).

# 4 Experiments

We test our algorithmic framework on randomly generated synthetic data and compare to other existing algorithms for SPCA. The code was implemented in Python using standard libraries.

We refer to our general algorithm from Section 3 that uses the $Q$ statistic as SPCAvSLR. For our SLR "black-box," we use thresholded Lasso [47].[8] (We experimented with other SLR algorithms such as the forward-backward algorithm of [46] and CoSaMP[9] [32], but results were similar and only slower.)

For more details on our experimental setup, including hyperparameter selection, see Appendix D.

**Support recovery** We randomly generate a spike $u \in \mathbb{R}^d$ by first choosing a random support of size $k$, and then using random signs for each coordinate (uniformity is to make sure Condition (C2) is met). Then spike is scaled appropriately with $\theta$ to build the spiked covariance matrix of our normal distribution, from which we draw samples.

We study how the performance of six algorithms vary over various values of $k$ for fixed $n$ and $d$.[10] As in the [15], our measure is the fraction of true support. We compare SPCAvSLR with the following algorithms: diagnoal thresholding, which is a simple baseline; "SPCA" (ZHT [49]) is a fast heuristic also based on the regression idea; the truncated power method of [45], which is known for both strong theoretical guarantees and empirical performance; covariance thresholding, which has state-of-the-art theoretical guarantees.

We modified each algorithm to return the top $k$ most likely coordinates in the support (rather than thresholding based on a cutoff); for algorithms that compute a candidate eigenvector, we choose the top $k$ coordinates largest in absolute value.

We observe that SPCAvSLR performs better than covariance thresholding and diagonal thresholding, but its performance falls short of that of the truncated power method and the heuristic algorithm of [49]. We suspect the playing with different SLR algorithms may slightly improve its performance. The reason for the gap between performance of SPCAvSLR and other state of the arts algorithms despite its theoretical guarantees is open to further investigation.

**Hypothesis testing** We generate data under two different distributions: for the spiked covariance model, we generate a spike $u$ by sampling a uniformly random direction from the $k$-dimensional unit sphere, and embedding the vector at a random subset of $k$ coordinates among $d$ coordinates; for the null, we draw from standard isotropic Gaussian. In a single trial, we draw $n$ samples from each distribution and we compute various statistics[11] (diagonal thresholding (DT), Minimal Dual Perturbation (MDP), and our $Q$ statistic, again using thresholded Lasso). We repeat for 100 trials, and plot the resulting empirical distribution for each statistic. We observe similar performance of

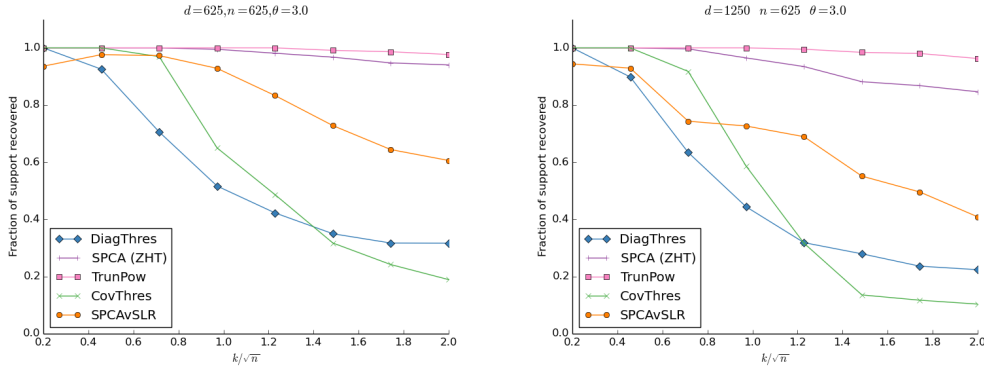

Figure 1: Performance of diagonal thresholding, SPCA (ZHT), truncated power method, covariance thresholding, and SPCAvSLR for support recovery at $n = d = 625$ (left) and $n = 625, d = 1250$ (right), varying values of $k$, and $\theta = 3.0$. On the horizontal axis we show $k/\sqrt{n}$; the vertical axis is the fraction of support correctly recovered. Each datapoint on the figure is averaged over 50 trials.

DT and $Q$, while MDP seems slightly more effective at distinguishing $H_0$ and $H_1$ at the same signal strength (that is, the distributions of the statistics under $H_0$ vs. $H_1$ are more well-separated).

**Rescaling variables** As discussed in Section 3.4, our algorithms are robust to rescaling the covariance matrix to the correlation matrix. As illustrated in Figure 2 (right), DT fails while $Q$ appears to be still effective for distinguishing hypotheses the same regime of parameters. Other methods such as MDP and CT also appear to be robust to such rescaling (not shown). This suggests that more modern algorithms for SPCA may be more appropriate than diagonal thresholding in practice, particularly on instances where the relative scales of the variables may not be accurate or knowable in advance, but we still want to be able to find correlation between the variables.

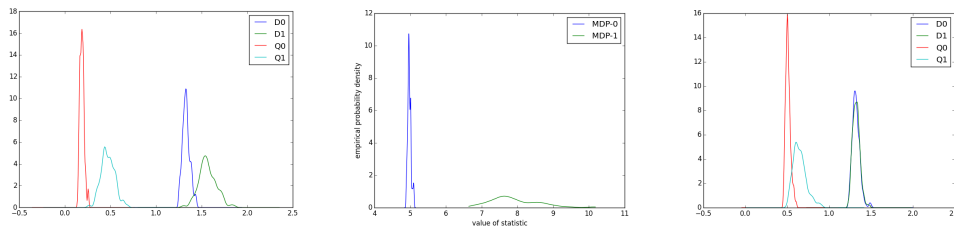

Figure 2: Performance of diagonal thresholding (D), MDP, and $Q$ for hypothesis testing at $n = 200, d = 500, k = 30, \theta = 4$ (left and center). T0 denotes the statistic $T$ under $H_0$, and similarly for T1. The effect of rescaling the covariance matrix to make variances indistinguishable is demonstrated (right).

## 5 Previous work

Here we discuss in more detail previous approaches to SPCA and how it relates to our work. Various approaches to SPCA have been designed in an extensive list of prior work. As we cannot cover all of them, we focus on works that aim to give computationally efficient (i.e. polynomial time) algorithms with provable guarantees in settings similar to ours.

These algorithms include fast, heuristic methods based on $\ell_1$ minimization [23, 49], rigorous but slow methods based on natural semidefinite program (SDP) relaxations [13, 1, 41, 44], iterative methods motivated by power methods for approximating eigenvectors [45, 24], non-iterative methods based on random projections [20], among others. Many iterative methods rely on initialization schemes, such as ordinary PCA or diagonal thresholding [22].

Below, we discuss the known sample bounds for support recovery and hypothesis testing.

**Support recovery** [1] analyzed both diagonal thresholding and an SDP for support recovery under the spiked covariance model.[12] They showed that the SDP requires an order of $k$ fewer samples when the SDP optimal solution is rank one. However, [27] showed that the rank one condition does not happen in general, particularly in the regime approaching the information theoretic limit ($\sqrt{n} \lesssim k \lesssim {}^n/\log d$). This is consistent with computational lower bounds from [3] ($k \gtrsim \sqrt{n}$), but a small gap remains (diagonal thresholding and SDP's succeed only up to $k \lesssim \sqrt{n/\log d}$). The above gap was closed by the covariance thresholding algorithm, first suggested by [27] and analyzed by [15], that succeeds in the regime $\sqrt{n/\log d} \lesssim k \lesssim \sqrt{n}$, although the theoretical guarantee is limited to the regime when $d/n \to \alpha$ due to relying on techniques from random matrix theory.

**Hypothesis testing** Some works [4, 1, 17] have focused on the problem of detection. In this case, [4] observed that it suffices to work with the much simpler dual of the standard SDP called Minimal Dual Perturbation (MDP). Diagonal thresholding (DT) and MDP work up to the same signal threshold $\theta$ as for support recovery, but MDP seems to outperform DT on simulated data [4]. MDP works at the same signal threshold as the standard SDP relaxation for SPCA. [17] analyze a statistic based on an SDP relaxation and its approximation ratio to the optimal statistic. In the regime where $k, n$ are proportional to $d$, their statistic succeeds at a signal threshold for $\theta$ that is independent of $d$, unlike the MDP. However, their statistic is quite slow to compute; runtime is at least a high order polynomial in $d$.

**Regression based approaches** To the best of our knowledge, our work is the first to give a general framework for SPCA that uses SLR in a *black-box* fashion. [49] uses specific algorithms for SLR such as Lasso as a subroutine, but they use a heuristic alternating minimization procedure to solve a non-convex problem, and hence lack any theoretical guarantees. [31] applies a regression based approach to a restricted class of graphical models. While our regression setup is similar, their statistic is different and their analysis depends directly on the particulars of Lasso. Further, their algorithm requires extraneous conditions on the data.[9] also uses a reduction to linear regression for their problem of sparse subspace estimation. Their iterative algorithm depends crucially on a good initialization done by a diagonal thresholding-like pre-processing step, which fails under rescaling of the data.[13] Furthermore, their framework uses regression for the specific case of orthogonal design, whereas our design matrix can be more general as long as it satisfies a condition similar to the restricted eigenvalue condition. On the other hand, their setup allows for more general $\ell_q$-based sparsity as well as the estimation of an entire subspace as opposed to a single component. [29] also achieves this more general setup, while still suffering from the same initialization problem.

**Sparse priors** Finally, connections between SPCA and SLR have been noted in the probabilistic setting [26, 25], albeit in an indirect manner: the same sparsity-inducing priors can be used for either problem. We view our work as entirely different as we focus on giving a black-box reduction. Furthermore, provable guarantees for the EM algorithm and variational methods are lacking in general, and it is not immediately obvious what signal threshold their algorithm achieves for the single spike covariance model.

# 6   Conclusion

We gave a black-box reduction for reducing instances of the SPCA problem under the spiked covariance model to instances of SLR. Given oracle access to SLR black-box meeting a certain natural condition, the reduction is shown to efficiently solve hypothesis testing and support recovery.

Several directions are open for future work. The work in this paper remains limited to the Gaussian setting and to the single spiked covariance model. Making the results more general would make the connection made here more appealing. Also, the algorithms designed here, though simple, seem a bit wasteful in that they do not aggregate information from different statistics. Designing a more efficient estimator that makes a more efficient use of samples would be interesting. Finally, there is certainly room for improvement by tuning the choice of the SLR black-box to make the algorithm more efficient for use in practice.

## Footnotes

[3]From here on, we will use "coordinate" and "variable" interchangeably.

[4]In the scaling limit $d/n \to \alpha$ as $d, n \to \infty$, the covariance thresholding algorithm [15] theoretically succeeds at a signal strength that is an order of $\sqrt{\log d}$ smaller. However, our experimental results indicate that with an appropriate choice of black-box, our Q algorithm outperforms covariance thresholding

[5]Solving SPCA based on the correlation matrix was suggested in a few earlier works [49, 41].

[6] By the theory of linear minimum mean-square-error (LMMSE) confirms that this choice of $\beta^*$ minimizes the error $\sigma^2$. See Appendix B.1, B.2 for details of this calculation.

[7] As pionted out by a reviewer, Note that this statistic is actually equivalent to $R^2$ up to rescaling by sample variance. Note that our formula is slightly different though as we use the sample variance computed with population mean as opposed to sample mean, as the mean is known to be zero.

[8]Thresholded Lasso is a variant of lasso where after running lasso, we keep the $k$ largest (in magnitude) coefficients to make the estimator k-sparse. Proposition 2 in [47] shows that thresholded Lasso satisfies Condition 2.2.

[9]CoSaMP in theory requires the stronger condition of restricted isometry on the "sensing" matrix.

[10]We remark that while the size of this dataset might seem too small to be representative "high-dimensional" setting, these are representative of the usual size of dataset that these methods are usually tested on. One bottleneck is the computation of the covariance matrix.

[11]While some of the algorithms used for support recovery in the previous section could in theory be adapted for hypothesis testing, the extensions were immediate so we do not consider them here.

[12]They analyze the subcase when the spike is uniform in all $k$ coordinates.

[13]See Section 3.4 for more discussion on rescaling.

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
