[Supplementary Material]

# Appendix

The appendix is organized as follows. In Section A, we give additionaal background on the literature for sparse linear regression. In Section B, we give calculations for quantities from Section 3.1 in the main text, where we set up our main linear model. In Section C, we give complete proofs for Theorems 3.1 and 3.2. Finally, we provide additional details for our experiments in Section D.

## A   Additional background on SLR

**Efficient methods** The $\ell_0$ estimator, which minimizes the reconstruction error $\|y - \mathbb{X}\widehat{\beta}\|_2^2$ over all $k$-sparse regression vectors, achieves prediction error bound of form ([7], [35]): $1/n\|\mathbb{X}\beta^* - \mathbb{X}\widehat{\beta}\|_2^2 \lesssim (\sigma^2 k \log d)/n$ but takes exponential time $O(n^k)$ to compute. Various efficient methods have been proposed to circumvent this computational intractability: basis pursuit, Lasso[38], and the Dantzig selector [10] are some of initial approaches. Greedy pursuit methods such as OMP [30], IHT[6], CoSaMP[32], and FoBa[46] among others offer more computationally efficient alternatives.[14] These algorithms achieve the same prediction error guarantee as $\ell_0$ up to a constant, but under the assumption that $\mathbb{X}$ satisfies certain properties, such as restricted eigenvalue ([5]), compatibility ([39]), restricted isometry property ([11]), (in)coherence ([8]), among others. In this work, we focus on the restricted eigenvalue (see Definition 2.1 for a formal definition). We remark that restricted eigenvalue is among the weakest, and is only slightly stronger than the compatibility condition. Moreover, [47] give complexity-theoretic evidence for the necessity of dependence on the RE constant for certain worst case instances of the design matrix. See [40] for implications between various conditions. Without such conditions on $\mathbb{X}$, the best known guarantees provably obtain only a $1/\sqrt{n}$ decay rather than a $1/n$ decay in prediction error as number of samples increase; [48] give some evidence that this gap may be unavoidable.

**Optimal estimators** The SLR estimators we consider are efficiently computable. Another line of work considers arbitrary estimators that are not necessarily efficiently computable. These include BIC [7], Exponential Screening [36], and Q-aggregation [12]. Such estimators achieve strong guarantees regarding minimax optimality in the form of oracle inequalities on MSE.

**Restricted Eigenvalue** The restricted eigenvalue (RE) lower bounds the quadratic form defined by $\mathbb{X}$ in all (approximately) sparse directions. RE is related to more general notions such as the *restricted strong convexity* [33], which roughly says that loss function is not too "flat" near the point of interest; this allows us to convert convergence in loss value to convergence in parameter value. In general when $d > n$, we cannot guarantee this for all directions, but it suffices to consider the set $\mathbb{C}(S)$ of "mostly" sparse directions.

We remark that the above condition is very natural and likely unavoidable. [47] indicate that the dependence of the above prediction error guarantee on RE cannot be removed, under a standard conjecture in complexity theory. [34, 37] show that RE holds with high probability for correlated Gaussian designs (though it remains NP-hard to verify it [2]).

A recent line of work [14, 19] studies the algorithmic hardness of SLR when $\mathbb{X}$ has Gaussian design.

## B   Deferred calculations from Section 3.1

### B.1   Linear minimum mean-square-error estimation

Given random variables $Y$ and $X$ (this can be a vector more generally), what is the best prediction for $Y$ conditioned on knowing $X = x$? What is considered "best" can vary, but here we consider the mean squared error. That is, we want to come up with $\hat{y}(x)$ s.t.

$$\mathbb{E}[(Y - \hat{y})^2]$$

is minimized.

It is not hard to show that $\hat{y}$ is just the conditional expectation of $Y$ conditioned on $X$. The minimum mean-square-error estimate can be a highly nontrivial function of $X$.

The *linear minimum mean-square-error* (LMMSE) estimate instead restricts the attention to estimators of the form $\hat{Y} = AX + b$. Notice here that $A$ and $b$ are fixed and are not functions of $X$.

One can show that the LMMSE estimator is given by: $A = (\Sigma_{XX})^{-1})\Sigma_{XY}$, where $\Sigma$. is the appropriately indexed covariance matrix, and $b$ is chosen in the obvious way to make our estimator unbiased.

## B.2 Calculations for the linear model

To recap our setup, we input the design matrix $\mathbb{X} = \mathbf{X}_{-i}$ and the response variable $y = \mathbf{X}_i$ as inputs to an SLR black-box. Our goal is to express $y$ as a linear function of $\mathbb{X}$ plus some independent noise $w$. Without loss of generality let $i = 1$, and for our discussion below assume $S = \{1, ..., k\}$. For illustration, at times we will simplify our calculation further for the uniform case where $u_i = \frac{1}{\sqrt{k}}$ for $1 \leq i \leq k$ and $u_i = 0$ for $i > k$.

For the moment, just consider one row of $\mathbf{X}$, corresponding to one particular sample $X$ of the original SPCA distribution. Since $X$ is jointly Gaussian, we can express (the expectation of) $y = X_1$ as a linear function of the other coordinates:

$$\mathbb{E}[X_1 | X_{2:d} = x_{2:d}] = \Sigma_{1,2:d}(\Sigma_{2:d})^{-1}x_{2:d}$$

Hence we can write

$$X_1 = \Sigma_{1,2:d}(\Sigma_{2:d})^{-1}X_{2:d} + w$$

where $w \sim \mathcal{N}(0, \sigma^2)$ for some $\sigma$ to be determined and $w \perp X_i$ for $i = 2, .., d$.

By directly computing the variance of the above expression for $X_1$, we deduce an expression for the noise level:

$$\sigma^2 = \Sigma_{11} - \Sigma_{1,2:d}(\Sigma_{2:d})^{-1}\Sigma_{2:d,1}$$

Note that $\sigma^2$ is just $\Sigma_{11}$ under $H_0$. We proceed to compute $\sigma^2$ under $H_1$, when $\Sigma = I_d + \theta uu^\top$. To compute $(\Sigma_{2:d})^{-1}$, we use (a special case of) the Sherman-Morrison formula: $(I + wv^\top)^{-1} = I - \frac{wv^\top}{1+v^\top w}$.

$$\Sigma_{2:d}^{-1} = \left(I_{d-1} + \theta u_{-1}u_{-1}^\top\right)^{-1} = I_{d-1} - \frac{\theta}{1 + (1 - u_1^2)\theta}u_{-1}u_{-1}^\top$$

where $u_{-1} \in \mathbb{R}^{d-1}$ is $u$ restricted to coordinates $2, ..., d$.

$$\Sigma_{1,2:d}(\Sigma_{2:d})^{-1}\Sigma_{2:d,1} = \left(\frac{\theta u_1}{1 + (1 - u_1^2)}\right)^2 u_{-1}^\top(I + \theta u_{-1}u_{-1}^\top)u_{-1}$$

$$= \frac{\theta^2 u_1^2(1 - u_1^2)}{1 + (1 - u_1^2)\theta}$$

(specializing to uniform case again)

$$= \frac{\theta^2}{k}\left(1 - \frac{1}{k}\right)\frac{1}{1 + \frac{k-1}{k}\theta} \approx \frac{\theta^2}{k(1 + \theta)}$$

Finally, substituting into the expression for $\sigma^2$

$$\sigma^2 = 1 + \theta u_1^2 - \frac{\theta^2 u_1^2(1 - u_1^2)}{1 + (1 - u_1^2)\theta}$$

$$= 1 + \frac{\theta u_1^2}{1 + (1 - u_1^2)\theta}$$

$$\leq 2 \quad \text{if } \theta \leq 1$$

We remark that the noise level of column 1 has been reduced by roughly $\tau := \frac{\theta^2}{k(1+\theta)}$ by regressing on correlated columns.

In summary, under $H_1$ (and if $1 \in S$) we can write

$$y = \mathbb{X}\beta^* + w$$

where

$$
\begin{aligned}
\beta^* &= (\Sigma_{2:d})^{-1}\Sigma_{2:d,1} \\
&= (I - \frac{\theta}{1 + (1-u_1)^2\theta}u_{-1}u_{-1}^\top)\theta u_1 u_{-1} \\
&= \theta u_1 \left(1 - \frac{\theta}{1 + (1-u_1^2)\theta}(1 - u_1^2)\right) u_{-1} \\
&= \frac{\theta u_1}{1 + (1-u_1^2)\theta}u_{-1}
\end{aligned}
$$

(technically, the definition of $\beta^*$ on the RHS is a $k-1$ dimensional vector, but we augment it with zeros to make it $d-1$ dimensional) and $w \sim \mathcal{N}(0, \sigma^2)$ where $\sigma^2 = 1 + \frac{\theta u_1^2}{1+(1-u_1^2)\theta}$. Note that in the uniform case, $\beta^* \to \frac{1}{k-1}\mathbf{1}_{k-1}$ as $\theta \to \infty$ where $\mathbf{1}_{k-1}$ is uniform 1 on first $k-1$ coordinates, as expected.

## B.3 Properties of the design matrix $\mathbb{X}$

**Restricted eigenvalue (RE)**  Here we check that $\mathbb{X}$ defined as in Section 3.1 has constant restricted eigenvalue constant. This allows us to apply Condition 2.2 for the SLR black-box with good guarantee on prediction error.

The rows of $\mathbb{X}$ are drawn from $\mathcal{N}(0, I_{d-1\times d-1} + \theta u_{-1}u_{-1}^\top)$ where $u_{-1}$ is $u$ restricted to coordinates $2, ..., d$ wlog.[15]

Let $\Sigma = I_{d-1\times d-1} + \theta u_{-1}u_{-1}^\top$. We can show that $\Sigma^{1/2}$ satisfies RE with $\gamma = 1$ by bounding $\Sigma$'s minimum eigenvalue. First, we compute the eigenvalues of $\theta u_{-1}u_{-1}^\top$. $\theta u_{-1}u_{-1}^\top$ has a nullspace of dimension $d-2$, so eigenvalue 0 has multiplicity $d-2$. $u_{-1}$ is a trivial eigenvector with eigenvalue $\theta u_{-1}^\top u_{-1} = \theta\frac{k-1}{k}$. Therefore, $\Sigma$ has eigenvalues 1 and $1 + \theta\frac{k-1}{k}$.

Now we can extend this to the sample matrix $\mathbb{X}$ by applying Corollary 1 of [34] (also see Example 3 therein), and conclude that as soon as $n \gtrsim \frac{\max_j \Sigma_{jj}}{\gamma^2}k\log d = C(1 + \frac{\theta}{k})k\log d = \Omega(k\log p)$ the matrix $\mathbb{X}$ satisfies RE with $\gamma(\mathbb{X}) = 1/8$.

We remark that the following small technical condition also appears in known bounds on prediction error:

**Column normalization**  This is a condition on the scale of $\mathbb{X}$ relative to the noise in SLR, which is always $\sigma^2$.

$$\frac{\|\mathbb{X}\theta\|_2^2}{n} \leq \|\theta\|_2^2$$

for all $\theta \in B_0(2k)$

We can always rescale $\mathbf{X}$ (and hence $\mathbb{X}$) to satisfy this, which would also rescale the noise level $\sigma$ in our linear model since the noise is derived from coming $\mathbf{X}$ from the SPCA generative model, rather than added independently as in the usual SLR setup.

Hence, since all scale dependent quantities are scaled by the same amount when we scale the original data $\mathbf{X}$, wlog we may continue to use the same $\mathbb{X}$ and $\sigma$ in our analysis. As the column normalization condition does not affect us, we drop it from Condition 2.2 of our black-box assumption.

## C Proofs of main Theorems

In this section we analyze the distribution of $Q_i$ under both $H_0$ and $H_1$ on our way to proving Theorems 3.1 and 3.2. Note that though the dimension and the sparsity of our SLR instances are $d-1$ and $k-1$ (since we remove one column from the SPCA data matrix $\mathbf{X}$ to obtain the design matrix $\mathbb{X}$), for ease of exposition we just use $d, k$ in their place since it does not affect the analysis in any meaningful way.

First, we review a useful tail bound on $\chi^2$ random variables.

**Lemma C.1** (Concentration on upper and lower tails of the $\chi^2$ distribution ([28], Lemma 1)). *Let $Z$ be the $\chi^2$ random variable with $k$ degrees of freedom. Then,*

$$\Pr(Z - k \geq 2\sqrt{kt} + 2t) \leq \exp(-t)$$
$$\Pr(Z - X \geq 2\sqrt{kt}) \leq \exp(-t)$$

We can simplify the upper tail bound as follows for convenience:

**Corollary C.2.** *For $\chi^2$ r.v. $Z$ with $k$ degrees of freedom and deviation $t \geq 1, \Pr\left(\frac{Z-k}{k} \geq 4t\right) \leq \exp(-kt)$.*

### C.1 Analysis of $Q_i$ under $H_1$

Without loss of generality assume the support of $u$, denoted $S$, is $\{1, ..., k\}$ and consider the first coordinate. We expand $Q_1$ by using $y = \mathbb{X}\beta^* + w$ as follows:

$$Q_1 = \frac{1}{n}\|y\|_2^2 - \frac{1}{n}\|y - \mathbb{X}\widehat{\beta}\|_2^2 = \frac{1}{n}\|\mathbb{X}\beta^* + w\|_2^2 - \frac{1}{n}\|\mathbb{X}\beta^* - \mathbb{X}\widehat{\beta}\|_2^2 - \frac{2}{n}w^\top(\mathbb{X}\beta^* - \mathbb{X}\widehat{\beta}) - \frac{1}{n}\|w\|_2^2$$

$$= \frac{1}{n}\|\mathbb{X}\beta^*\|_2^2 - \frac{2}{n}w^\top \mathbb{X}\beta^* - \frac{1}{n}(\|\mathbb{X}\beta^* - \mathbb{X}\widehat{\beta}\|_2^2) - \frac{2}{n}w^\top(\mathbb{X}\beta^* - \mathbb{X}\widehat{\beta})$$

Observe that the noise term $\|w\|_2^2$ cancels conveniently.

Before bounding each of these four terms, we introduce a useful lemma to bound cross terms involving noise $w$:

**Lemma C.3** (Lemmas 8 and 9, [35]). *For any fixed $\mathbb{X} \in \mathbb{R}^{n \times d}$ and independent noise vector $w \in \mathbb{R}^n$ with i.i.d. $\mathcal{N}(0, \sigma^2)$ entries:*

$$\frac{|w^\top \mathbb{X}\theta|}{n} \leq 9\sigma \frac{\|\mathbb{X}\theta\|_2}{n}\sqrt{k \log \frac{d}{k}}$$

*for all $\theta \in B_0(2k)$ w.p. at least $\geq 1 - 2\exp(-40k \log(d/k))$*

We bound each term as follows:

*Term 1.* The first term $\frac{\|\mathbb{X}\beta^*\|_2^2}{n}$ contains the signal from the spike; notice its resemblance to the $k$-sparse eigenvalue statistic. Rewritten in another way,

$$(\beta^*)^\top \frac{\mathbb{X}^\top \mathbb{X}}{n}\beta^* = (\beta^*)^\top \widehat{\Sigma}_{2:d}\beta^*$$

Hence, we expect this to concentrate around $(\beta^*)^\top \Sigma_{2:d}\beta^*$, which simplifies to (see Appendix B.2 for the full calculation):

$$(\beta^*)^\top \Sigma_{2:d}\beta^* = (\Sigma_{1,2:d}\Sigma_{2:d}^{-1})\Sigma_{2:d}(\Sigma_{2:d}^{-1}\Sigma_{2:d,1}) = \frac{\theta^2 u_1^2(1 - u_1^2)}{1 + (1 - u_1^2)\theta}$$

For concentration, observe that we may rewrite

$$(\beta^*)^\top \widehat{\Sigma}_{2:d}\beta^* = \frac{1}{n}\sum_{i=1}^n (\mathbb{X}^{(i)}\beta^*)^2$$

where $\mathbb{X}^{(i)}$ is the $i$th row, representing the $i$th sample. This is just an appropriately scaled chi-squared random variable with $n$ degrees of freedom (since each $\mathbb{X}^{(i)}\beta^*$ is i.i.d. normal), and the expected

value of each term in the sum is the same as computed above. Applying a lower tail bound on $\chi^2$ distribution (see Appendix ), with probability at least $1 - \delta$ we have

$$(\beta^*)^\top \widehat{\Sigma}_{2:d}\beta^* \geq \frac{\theta^2 u_1^2(1 - u_1^2)}{1 + (1 - u_1^2)\theta} \cdot \left(1 - 2\sqrt{\frac{\log(1/\delta)}{n}}\right)$$

Choosing $\delta = \exp(-k \log d)$,

$$
\begin{aligned}
\frac{\|\mathbb{X}\beta^*\|_2^2}{n} &\geq \frac{\theta^2 u_1^2(1 - u_1^2)}{1 + (1 - u_1^2)\theta} \cdot \left(1 - 2\sqrt{\frac{k \log d}{n}}\right) \\
&\overset{(a)}{\geq} \frac{1}{2} \cdot \frac{\theta^2 u_1^2(1 - u_1^2)}{1 + (1 - u_1^2)\theta} \\
&\overset{(b)}{\geq} \frac{c_{min}^2}{4}\frac{\theta^2}{k}
\end{aligned}
\tag{1}
$$

where $(a)$ as long as $n > 16k \log d$ and $(b)$ since $\theta \leq 1$ and $u_1^2(1 - u_1^2) \gtrsim c_{min}^2/k$ under Condition (C1).

*Term 2.* The absolute value of the second term $\frac{2}{n}w^\top \mathbb{X}\beta^*$ can be bounded by $18\frac{\|\mathbb{X}\beta^*\|_2}{n}\sqrt{k \log \frac{d}{k}}$ using Lemma C.3. From (1) as long as $\theta^2 > \frac{c_1}{c_{min}^2}\frac{k^2 \log d}{n}$ ($c_1$ is some constant that we will choose later),

$$\frac{\|\mathbb{X}\beta^*\|_2^2}{n} \geq \frac{c_{min}^2}{4}\frac{\theta^2}{k} \geq \frac{c_1}{4}\frac{k \log d}{n}$$

so the first two terms together are lower bounded by:

$$\frac{\|\mathbb{X}\beta^*\|_2}{n}\left(\|\mathbb{X}\beta^*\|_2 - 18\sqrt{k \log d/k}\right) \geq \frac{c_1}{5}\frac{k \log d}{n}, \tag{2}$$

for large enough constant $c_1$.

*Term 3.* The third term, which is the prediction error $\frac{\|\mathbb{X}\beta^* - \mathbb{X}\widehat{\beta}\|_2^2}{n}$, is upper bounded by $\frac{C}{\gamma(X)^2}\frac{\sigma^2 k \log d}{n}$ with probability at least $1 - C\exp(-C'k \log d)$ by Condition 2.2 on our SLR black-box. Note $\sigma^2 < 2$ if we assume $\theta \leq 1$.[16] Now, $\gamma(X) \geq \frac{1}{8}$ with probability at least $1 - C\exp(-C'n)$ if $n > C''k \log d$ since $\theta \leq 1$ (see Appendix B.3 for more details). Then,

$$\frac{1}{n}\|\mathbb{X}\beta^* - \mathbb{X}\widehat{\beta}\|_2^2 \leq C\frac{k \log d}{n}$$

*Term 4.* The contribution of the last cross term $\frac{2}{n}w^\top \mathbb{X}(\beta^* - \widehat{\beta})$ can also be bounded by Lemma C.3 w.h.p. (note $\beta^* - \widehat{\beta} \in B_0(2k)$)

$$\frac{|w^\top \mathbb{X}(\beta^* - \widehat{\beta})|}{n} \leq 9\sigma \frac{\|\mathbb{X}(\beta^* - \widehat{\beta})\|_2}{n}\sqrt{k \log \frac{d}{k}}.$$

Combined with the above bound for prediction error, this bounds the cross term's contribution by at most $C\frac{k \log d}{n}$.

Putting the bounds on four terms together, we get the following lower bound on $Q$.

**Lemma C.4.** *There exists constants $c_1, c_2, c_3, c_4$ s.t. if $\theta^2 > \frac{c_1}{c_{min}^2}\frac{k^2 \log d}{n}$ and $n > c_2 k \log d$, with probability at least $1 - c_3 \exp(-c_4 k \log d)$, for any $i \in S$ that satisfies the size bound in Condition (C1),*

$$Q_i > \frac{13k \log d}{n}$$

*Proof.* From 1-4 above, by union bound, all four bounds fail to hold with probability at most $c_3 \exp(-c_4 k \log d)$ for appropriate constants if $\theta^2 > \frac{c_1}{c_{min}^2} \frac{k^2 \log d}{n}$ (required by Term 2) and $n > c_2 k \log d$ for some $c_2 > 0$ (note that both terms 1 and 3 require sufficient number of samples $n$). If all four bounds hold, we have:

$$Q_i > \frac{c_1}{5} \frac{k \log d}{n} - C' \frac{k \log d}{n}$$

where $C, C'$ are just some constants. So if $c_1$ is sufficiently large, the above bound is greater than $\frac{13k \log d}{n}$.[17] $\qquad \square$

## C.2 Analysis of $Q_i$ under $H_0$

We could proceed by decomposing $Q_i$ the same way as in $H_1$; all the error terms including prediction error are still bounded by $O(k \log d / n)$ in magnitude, and the signal term is gone now since $\beta^* = 0$. This will give the same upper bound (up to a constant) as the following proof is about to show. However, we find the following direct analysis more informative and intuitive.

Since our goal is to upper bound $Q_i$ under $H_0$, we may let $\widehat{\beta}$ be the optimal possible choice given $y$ and $\mathbb{X}$ (one that minimizes $\|y - \mathbb{X}\widehat{\beta}\|_2^2$, and hence maximizes $Q_i$). We further break this into two steps. We enumerate over all possible subsets $S$ of size $k$, and conditioned on each $S$, choose the optimal $\widehat{\beta}$.

Fix some support $S$ of size $k$. The span of $\mathbb{X}_S$ is at most a $k$-dimensional subspace of $\mathbb{R}^n$. Hence, we can consider some unitary transformation $U$ of $\mathbb{R}^n$ that maps the span of $\mathbb{X}_S$ into the subspace spanned by the first $k$ standard basis vectors. Since $U$ is an isometry by definition,

$$nQ_i = \|y\|_2^2 - \|y - \mathbb{X}\widehat{\beta}_S\|_2^2 = \|Uy\|_2^2 - \|Uy - U\mathbb{X}\widehat{\beta}_S\|_2^2$$

Let $\tilde{y} = Uy$. Since $U\mathbb{X}\widehat{\beta}_S$ has nonzero entries only in the first $k$ coordinates, the optimal choice (in the sense of maximizing the above quantity) of $\widehat{\beta}_S$ is to choose linear combinations of the first $k$ columns of $\mathbb{X}$ so that $U\mathbb{X}\widehat{\beta}_S$ equals the first $k$ coordinates of $\tilde{y}$. Then, $nQ_i$ is just the squared norm of the first $k$ coordinates of $\tilde{y}$. Since $U$ is some unitary matrix that is independent of $y$ (being a function of $\mathbb{X}_S$ which is independent of $y$), $\tilde{y}$ still has i.i.d. $\mathcal{N}(0,1)$ entries, and hence $nQ_i$ is a $\chi^2$-var with $k$ degrees of freedom.

Now we apply an upper tail bound on the $\chi^2$ distribution (see Corollary C.2). Choosing $t = 3 \log \frac{d}{k}$, and after union bounding over all $\binom{d}{k} \leq \left(\frac{de}{k}\right)^k$ supports $S$, $nQ_i > k + 12k \log \frac{d}{k} \geq 13k \log \frac{d}{k}$ with probability at most $\exp(-3k \log \frac{d}{k} + k \log \frac{de}{k}) \leq \exp(-k \log \frac{d}{k})$ as long as $\frac{d}{k} \geq e$.

**Lemma C.5.** *Under $H_0$, $\forall i$ $Q_i \leq \frac{13k \log \frac{d}{k}}{n}$ w.p. at least $1 - \exp(-k \log \frac{d}{k})$.*

**Remark C.6.** Union bounding over all $S$ is necessary for the analysis. For instance, we cannot just fix $S$ to be $S(\widehat{\beta})$ (this denotes the support of $\widehat{\beta}$) since $\widehat{\beta}$ is a function of $y$, so fixing $S$ changes the distribution of $y$.

**Remark C.7.** Observe that this analysis of $Q_i$ for $H_0$ also extends immediately to $H_1$ when coordinate $i$ is outside the support. The reason the analysis cannot extend to when $i \in S$ is because $U$ is not independent of $y$ in this case.

**Corollary C.8.** *Under $H_1$, if $i \notin S$, $Q_i \leq \frac{13k \log \frac{d}{k}}{n}$ w.p. at least $1 - \exp(-k \log \frac{d}{k})$.*

## C.3 Proof of Theorem 3.1

*Proof.* Proof follows immediately from Lemma C.4 and Lemma C.5. We can use our statistics $Q_i$ to separate $H_0$ and $H_1$. Under $H_0$, applying Lemma C.5 to each coordinate $i$ and union bounding, $\forall i$, $Q_i \leq \frac{13k \log \frac{d}{k}}{n}$ with probability at least $1 - \exp(-Ck \log d)$. Meanwhile, under $H_1$, if we consider

any coordinate $i$ that satisfies Condition (C1), Lemma C.4 gives

$$Q_i > \frac{13k \log d}{n}$$

with probability at least $1 - c_3 \exp(-c_4 k \log d)$. Since $\psi$ tests whether $Q_i > \frac{13k \log \frac{d}{k}}{n}$ for at least one $i$, $\psi$ distinguishes $H_0$ and $H_1$ successfully, with bound on type I and type II error probability $c_3 \exp(-c_4 k \log d)$ for appropriate constants $c_3, c_4$ (note, these may be different from those of Lemma C.4). For runtime, note that we make $d$ calls to the SLR black-box and work with matrices of size $n \times d$. $\qquad\square$

### C.4 Proof of Theorem 3.2

*Proof.* As long as every $u_i$ for $i \in S$ has magnitude $c_{min}/\sqrt{k}$ as in Condition (C2), we can repeat the same analysis from above to all coordinates in the support. If $\theta$ meets the same threshold, $Q_i > 13k \log \frac{d}{k}/n$ for all $i \in S$ with probability at least $1 - C \exp(-C'k \log d)$ by union bound. Also, recall $Q_i > 13k \log \frac{d}{k}/n$ for any $i \notin S$ with probability at most $C \exp(-C'k \log d)$ by Corollary C.8. By union bound over all $d - k$ coordinates outside the support, the error probability is at most $d \cdot C \exp(-C'k \log d) \le C \exp(-C''k \log d)$. We showed that with high probability we exactly recover the support $S$ of $u$.

Runtime analysis is identical to that for the hypothesis test. $\qquad\square$

**Running time** The runtime of both Algorithms 1 and 2 is $\tilde{O}(nd^2)$,[18] if we assume the SLR black-box takes nearly linear time in input size, $\tilde{O}(nd)$, which is achieved by known existing algorithms. Note that computing the sample covariance matrix alone takes $O(nd^2)$ time, assuming one is using a naive implementation of matrix multiplication. For a broad comparison, we consider spectral methods and SDP-based methods, though there are methods that do not fall in either category. Spectral methods such as covariance thresholding or truncated power method have an iteration cost $O(d^2)$ due to operating on $d \times d$ matrices, and hence have total running time $\tilde{O}(d^2)$ ($\tilde{O}(\cdot)$ hiding precise convergence rate) in addition to the same $O(nd^2)$ initialization time. SDP-based methods in general take $\tilde{O}(d^3)$ time, the time taken by interior point methods to optimize. So overall, Algorithms 1 and 2 are competitive choices for (single spiked) SPCA, at least theoretically, though they seem slower in practice.

## D  Experimental Setup

We provide some futher details of the experimental setup, including selection of hyperparameters.

For the "SPCA" algorithm of [49], we used their direct implementation using an initialization with PCA (rather than the self-contained alternating minimization algorithm they present as an alternative). We also leave out the $\ell_2$ ridge penalty for convenience of implementation (their algorithm already performed very well in our experiments, so it was unnecessary to implement the full version).

For the truncated power method, we used the convergence criterion that the $l_2$ norm of the difference between eigenvectors from two consecutive iterations is less than $\epsilon = 0.01$.

For covariance thresholding, we tried various levels of parameter $\tau$, which controls the threshold for soft-thresholding, and indeed it performed best at [15]'s recommended value of $\tau \approx 4$, which is the choice compared against.

For our $Q$-based algorithm SPCAvSLR, we used thresholded Lasso with $\lambda = 0.1$ where $\lambda$ controls the weight of the $\ell_1$ regularization. This is close to the recommended choice of $\lambda = 4\sigma\sqrt{\frac{\log d}{n}}$ from [47] for our parameter setting.

## Footnotes

[14]Note that some of these algorithms were presented for compressed sensing; nonetheless, their guarantees can be converted appropriately.

[15]We assume here that $1 \in S$ as in the previous section

[16] As smaller $\theta$ makes the problem only harder, we assume $\theta \leq 1$ for ease of computation and as standard in literature.

[17]The choice of constant 13 may seem a little arbitrary, but this is just to be consistent with Lemma C.5. There, the constant just falls out of convenient choices for simplification, and is not optimized for in particular.

[18]In what follows $\tilde{O}(\cdot)$ hides possible log and accuracy parameter $\varepsilon$ factors.