[Reviews · NeurIPS 2018]

Reviewer 1



The paper proposes an approach to reduce solving a special sparse PCA to a sparse linear regression (SLR) problem (treated as a black-box solution). It uses the spiked covariance model [17] and assumes that the number of nonzero components of the direction (u) is known, plus some technical conditions such as a restricted eigenvalue property. The authors propose algorithms for both hypothesis testing and support recovery, as well as provide theoretical performance guarantees for them. Finally, the paper argues that the approach is robust to rescaling and presents some numerical experiments comparing two variants of the method (based on SLR methods FoBa and LASSO) with two alternatives (diagonal thresholding and covariance thresholding). Strengths: - The addressed problem (sparse PCA) is interesting and important. - The paper is well-written, the general idea is relatively simple and easy to understand (the implementation of Algorithms 1 and 2 also looks straightforward). - As the underlying sparse linear regression (SLR) approach is treated as a black-box (as long as it satisfies Condition 2.4), the approach has several variants depending on the SLR used. - Both hypothesis testing and support recovery are addressed and the paper provides theoretical performance guarantees for them. - The authors also present experimental comparisons to demonstrate the advantages of the method over some previous solutions. Weaknesses: - It is obvious that there is a strong connection between PCA and linear regression (see for example [41]), thus the originality of the general idea is limited. - The spiked covariance model for PCA is quite restrictive (for example, Gaussian white noise) and unlikely to be met in practical applications. In fact Gaussianity is crucial for this approach, which makes the results less significant. - The assumption that the number of nonzero components of u are known is strong (though the authors argue that there are adaptive methods to adjust this). - It is not clear how easy it is to check / verify for a particular problem and SLR method that Condition 2.4 holds. It looks like a strong assumptions which is hard to verify. Ideally, the authors should have demonstrated verifying this condition on an example. - The fact that the underlying SLR method is treated as a black-box might hide the problem of selecting the appropriate SLR method. Which one should be used for various problems? Minor comments: - The authors argue that though random design matrices for linear regression can arise, it makes no difference to assume that it is deterministic (by conditioning). This argument is a bit misleading, as it is only true if the design matrix (\mathbb{X}) and the noise vector (w) affecting the observations are independent. This is not the case, for example, if the linear regression problem arises from a time-series problem, such as estimating the parameters of an autoregressive model (in which case the design matrix cannot be assumed to be deterministic). - It is not a good practice to cite works that are only available on arXiv (as they did not go through any review process, they could contain unsupported claims, etc.). Post-rebuttal comments: Thank you for your replies. It is a nice feature that the presented approach can turn a black-box SLR to a SPCA solver. Nevertheless, it requires strong assumptions, for example, the spiked covariance model, knowledge of sparsity, and RE, which limit its theoretical and practical relevance. I understand that these assumptions could be relaxed a bit, for example, the method could be extended to sub-Gaussian variables and unknown sparsity could be handled by binary search (for hypothesis testing). It would be good to discuss these issues in more detail in the revised paper.

Reviewer 2



The paper suggests a general framework for implementing sparse PCA based on sparse regression. The focus of the paper is theoretical, with the main emphasis on provable guarantees for the spiked covariance model. The paper shows that using black-box lasso solver, their method performs better than "covariance thresholding" (NIPS 2014). The paper is clearly written and appears to be sound. I vote for accepting. Caveat: I did not study the mathematical details of the proofs. MAJOR ISSUES 1. There appears to be some disconnect in the sparse PCA literature between "more applied papers" focusing on possible applications (e.g. Zou et al. 2006) and "more theoretical papers" focusing on theoretical guarantees using the spiked covariance model (e.g. diagonal thresholding, covariance thresholding etc.). Personally, I am not very familiar with the latter line of work. Despite a decent literature overview in Section 1.1, I am left a bit wondering about the relationship. (i) Given that Zou et al. is by far the most cited and well-known paper on sparse PCA (almost 2k citations on Google Scholar), would it make sense to add it to the comparison in Figure 1? I have literally no idea how it will perform in comparison to the methods analyzed here. I do understand that Zou et al. did not provide any theoretical guarantees. But one can surely still assess how their method works in this particular experiment? (ii) In a purely applied setting, when working with p>>n dataset (e.g. gene expression data) that is clearly very far away from spiked covariance model, what would the authors recommend? Using their method? Or using something like Zou et al.? Why? I would like to see some brief discussion of these questions. And what about p

Reviewer 3



# Summary: The authors consider the algorithmic relationship between spare linear regression (SLR) and sparse principal component analysis (SPCA). These relationships are subsequently used demonstrate how SLR methods may be used to solve the SPCA problem. The latter is a valuable tool in for interpretable data exploration and many methods have been proposed. The authors consider two problems: recovering the leading sparse PC and hypothesis testing (testing for non-isotropic noise). The manuscript is very well written. The extensive literature on both SLR and SPCA is well documented. The proposed algorithm and intuition behind it are also clearly communicated. The only reservations concern the experimental results, which are not extensive. My concerns are as follows: - the data generation method is unclear. - many of the hyper-parameters employed are not justified (they should at least be tuned in some data driven manner). For example, $\lambda=.1$ is used for the Lasso (line 298), but how such a value is selected is unclear. - Figures 1 and 2 are unclear. The x and y axis label is too small to read and no error bars are provided. # Minor comments: - line 218: LMMSE introduced but only defined later on (line 236)